# Electrochemical Urea Oxidation on Porous Ni and Ni–M (M = Ir, Pt) Electrodes Obtained via Molten-Salt Treatment Technique

**DOI:** 10.3390/ma18225069

**Published:** 2025-11-07

**Authors:** Dawid Kutyła, Michihisa Fukumoto, Hiroki Takahashi, Ryuu Takahashi, Katarzyna Skibińska, Piotr Żabiński

**Affiliations:** 1Faculty of Non-Ferrous Metals, AGH University of Kraków, al. A. Mickiewicza 30, 30-059 Kraków, Poland; kskib@agh.edu.pl (K.S.); zabinski@agh.edu.pl (P.Ż.); 2Graduate School of Engineering Science, Akita University, 1-1 Tegata-Gakuen-machi, Akita 010-8502, Japan; tkhshrk@gipc.akita-u.ac.jp (H.T.); m8024212@s.akita-u.ac.jp (R.T.)

**Keywords:** porous Ni-Pt foam, molten-salt deposition, alkaline electrolysis, HER, UOR

## Abstract

Porous Ni, Ni–Ir, and Ni–Pt electrodes were prepared on Ni substrates by molten-salt Al co-deposition followed by dealloying. SEM/EDS and XRD confirmed a Raney-type porous network with Ir or Pt present across the layer. A urea oxidation reaction (UOR) was tested in 1 M NaOH + 0.33 M urea by cyclic voltammetry and chronoamperometry at +0.40 V vs. SCE (60 min). Smooth Ni showed near-zero current. Porous Ni resulted in ~11 mA cm^−2^ initially and ~9 mA cm^−2^ after 60 min. Porous Ni–Ir started at ~7 mA cm^−2^ and fell to ~2 mA cm^−2^ within 5 min, indicating fast deactivation, likely due to Ir-oxide formation that suppresses the Ni^2+^/Ni^3+^ redox couple. Porous Ni–Pt remained at ~11 mA cm^−2^ over 60 min, consistent with a stable Ni–Pt effect in which Pt aids urea adsorption/activation while Ni provides the redox path for oxidation. Overall, Pt improves UOR performance, whereas Ir lowers it under these conditions.

## 1. Introduction

Hydrogen production by alkaline water electrolysis is a key pathway toward sustainable energy [1,2]. However, the overall efficiency of this process is limited by the sluggish anodic oxygen evolution reaction (OER), which requires a high overpotential and thus causes substantial energy losses [3]. One promising approach to reduce the energy demand is to replace the OER with the urea oxidation reaction (UOR) at the anode [4,5]. UOR occurs at a much lower thermodynamic potential than OER, which can significantly decrease the required cell voltage. Moreover, using urea (a common component of wastewater) as the fuel for the anodic reaction enables simultaneous hydrogen production and wastewater treatment. This dual benefit makes UOR-coupled electrolysis an attractive strategy for sustainable hydrogen generation.

Nickel (Ni) and its alloys are widely used as electrodes in alkaline electrolysis due to Ni’s good corrosion resistance and its ability to form catalytically active nickel oxyhydroxide (NiOOH) species in alkaline media [6,7]. The NiOOH/Ni(OH)_2_ redox couple on Ni surfaces plays a crucial role in facilitating OER and UOR by providing active sites for the oxidation of water or urea [8,9]. Nonetheless, pure nickel electrodes still suffer from considerable overpotentials, so various surface modification strategies have been explored to enhance their catalytic performance [10,11]. Two general approaches have been effective: (i) alloying or incorporating small amounts of noble metals (such as Pt, Pd, Ir, or Rh) to introduce highly active catalytic sites and (ii) increasing the electrochemically active surface area through nanostructuring or forming porous architectures [12,13,14,15]. Both approaches can markedly improve activity and durability by either lowering reaction activation barriers (in the case of noble-metal catalysts) or providing more reaction sites and better mass transport (in the case of porous structures) [16].

Among the methods to create high-surface-area Ni electrodes, molten-salt-assisted aluminum deposition and dissolution have emerged as a versatile route to fabricate porous Ni-based electrodes with a bicontinuous ligament network [17,18]. In this process, aluminum is electrodeposited into the Ni substrate at a high temperature (in a molten-salt medium), where it alloys with Ni, and is then selectively removed (dealloyed) by an anodic dissolution [19]. The result is a sponge-like Ni-rich structure with a network of interwoven ligaments and pores [20]. This porous structure provides an enormous surface area and also allows tuning of surface composition and oxide states via the alloying–dealloying process [21]. Such molten-salt Al treatment has been successfully applied to produce Raney-type Ni catalysts and other Ni alloy electrodes, yielding thick (~tens of micrometers) porous layers that significantly enhance electrode performance [22].

Previous studies have shown that incorporating noble metals like platinum into Ni-based porous electrodes can greatly boost their catalytic activity [23]. For example, Ni–Pt porous electrodes produced by molten-salt Al co-deposition and dealloying exhibited exceptional performance toward both the hydrogen evolution reaction (HER) at the cathode and OER at the anode. This improvement was attributed to the presence of Pt, which is a highly active catalyst for HER, as well as the unique restructuring that occurs during dealloying: Pt tends to redistribute and remain on the surface as aluminum is removed, continually re-exposing fresh active sites [24]. This means that the porous Ni–Pt electrode maintains a high density of accessible catalytic sites even after extensive use. Given these promising results with Pt, it is natural to ask whether other noble metals could similarly enhance Ni-based porous electrodes.

Iridium (Ir) is conventionally regarded as one of the most stable and OER-active noble metals (with IrO_2_ being a state-of-the-art OER catalyst). One might expect that introducing Ir into a Ni electrode could improve anodic reaction rates or stability. However, the behavior of Ir in the context of Al-modified porous Ni systems has not been fully explored. Unlike Pt, Ir has a strong affinity for aluminum: based on Al–Ir binary phase diagrams and prior research on Ni–Ir–Al systems, Ir readily forms stable aluminide intermetallic compounds such as Al_9_Ir_2_ and AlIr [25]. These Ir–Al phases are thermodynamically favored and could immobilize Ir in electrochemically inert configurations (that is, Ir bound in these intermetallic compounds might not be available on the surface to act as an active catalyst). If most of the Ir added to Ni ends up locked in such inactive phases after the molten-salt deposition/dissolution process, it would not contribute to catalysis. Thus, there is a clear need to clarify how Ir behaves in the porous Ni electrode prepared by the Al alloying and dealloying route, and whether any of the supposed benefits of Ir (for OER or UOR) can be realized in practice.

The present study addresses this knowledge gap by fabricating Ni–Ir porous electrodes via high-temperature molten-salt Al co-deposition and dealloying and systematically evaluating their structure and performance in comparison with analogous porous Ni and Ni–Pt electrodes. We first prepare a uniform Ir coating on Ni (to serve as a source of Ir during the molten-salt treatment) and then carry out Al deposition and dissolution in molten salts to create a porous structure. The resulting electrodes are thoroughly characterized: scanning electron microscopy (SEM) (JEOL 6000, Tokyo, Japan) with energy-dispersive X-ray spectroscopy (EDS) is used to observe surface morphology and cross-sectional elemental distribution, and X-ray diffraction (XRD) (Rigaku Miniflex, Tokyo, Japan) is performed to identify the crystalline phases present before and after the molten-salt process. Finally, electrochemical tests (Biologic SP-200, Seyssinet-Pariset, France) are conducted in an alkaline solution with urea to assess the catalytic activity for hydrogen evolution and urea oxidation. Cyclic voltammetry (CV) provides insight into the HER and UOR behavior of each electrode, and chronoamperometry (CA) at a fixed potential is used to evaluate the long-term stability and steady-state performance during UOR. Through this comprehensive comparison, we elucidate the role of Ir in the porous Ni framework and determine whether it offers any advantages over porous Ni alone or a porous Ni–Pt benchmark for the coupled hydrogen production and urea oxidation process.

## 2. Materials and Methods

### 2.1. Electrode Preparation

Flat nickel plates (99.9% Ni, ~1.5 cm^2^ geometric area each) were used as the base substrate for all electrodes. Prior to any coating, the Ni plates were mechanically polished with 800-grit SiC paper to ensure a clean, fresh surface and then thoroughly rinsed with acetone, ethanol, and deionized water and dried.

For Ni–Ir electrode fabrication, a thin layer of iridium was first electrodeposited onto the Ni substrate from an aqueous Ir plating bath. The galvanostatic deposition was carried out at a constant current density of 5 mA·cm^−2^ for 60 min, using a platinum counter electrode and an Ag/AgCl reference electrode. This procedure produced a continuous Ir coating over the Ni plate. The success of the Ir plating was evidenced by a uniform grayish metallic sheen on the Ni surface. A uniform noble-metal pre-layer is important, as it ensures that, in the next step (molten-salt alloying), the added Al will interact evenly with Ir across the entire surface rather than only in isolated spots. After preparing the Ir-coated Ni, the samples were subjected to a molten-salt treatment to deposit aluminum and then remove it to form a porous structure. The molten-salt melt was a eutectic mixture of NaCl–KCl containing 3.5 mol% AlF_3_, which was maintained at 750 °C under an inert argon atmosphere. Each Ir-coated Ni sample was immersed in the molten salt, and aluminum was electrodeposited onto it at a constant potential of −1.4 V vs. Ag/AgCl. The deposition was continued for 3600 s (1 h), during which Al^+3^ from the melt was reduced to Al^0^, which infiltrated the Ni/Ir surface (note: at 750 °C, deposited aluminum is liquid and can diffuse into the Ni matrix, forming Ni–Al and possibly Ir–Al intermetallic compounds). Immediately after this cathodic Al deposition, the electrode was subjected in situ to an anodic dissolution step: the potential was reversed to about −0.5 V vs. Ag/AgCl (still at 750 °C in the melt) and held until the anodic current decayed to near zero. This anodic polarization causes the freshly deposited aluminum (and Al-rich phases) to oxidize and dissolve back into the molten salt, selectively leaching out Al from the alloyed layer. Essentially, the Ni–Al (and any Ir–Al) compounds that formed are partially dealloyed, leaving behind a porous, nickel-rich skeleton. Once the current had dropped, indicating that most removable Al had been dissolved away, the sample was carefully withdrawn from the melt and allowed to air-cool to room temperature. Finally, any residual salt adhering to the sample was removed by rinsing in hot deionized water, yielding the finished porous Ni–Ir electrode.

For comparison, two other types of electrodes were prepared using similar procedures: porous Ni (without any Ir or other noble metal, obtained by performing the same molten-salt Al deposition/dissolution on a bare Ni plate) and porous Ni–Pt (obtained by plating a thin layer of Pt onto Ni instead of Ir and then carrying out the identical molten-salt Al treatment). The porous Ni–Pt electrode serves as a benchmark, since Pt is known to enhance catalytic performance, and such electrodes have shown excellent results in earlier studies. All samples (smooth Ni, porous Ni, porous Ni–Ir, and porous Ni–Pt) had the same geometric area (~1.5 cm^2^) for fair comparison.

### 2.2. Electrode Characterization

The morphology and composition of the electrodes were characterized using scanning electron microscopy and X-ray diffraction. Surface and cross-sectional microstructures were observed with a JEOL (Tokyo, Japan) field-emission SEM operated at appropriate accelerating voltages to resolve fine features. To examine the distribution of elements (Ni, Ir, and Al) across the structure, the SEM was equipped with an energy-dispersive X-ray spectroscopy detector for elemental mapping. Cross-sections of the porous electrodes were prepared by cutting the sample and gently polishing the cross-section (using fine polish and a diamond suspension) to reveal internal features without disturbing the porous architecture. These cross-sections were then analyzed by EDS mapping to see how Al and Ir were distributed after the molten-salt process.

Crystal phase identification was performed by X-ray diffraction (XRD) using Cu Kα radiation (λ ≈ 1.54 Å). XRD patterns were collected in the 2θ range of approximately 20–80° (with a step size of around 0.02° and appropriate counting time per step). Each sample was examined at three stages: (i) the pure Ni substrate (for reference), (ii) after Ir electroplating but before molten-salt treatment, and (iii) after the full molten-salt Al deposition/dissolution process. The diffraction peaks were compared with standard reference patterns (e.g., from the ICDD database) to identify the phases present. This allowed us to detect the formation of any Ni–Al intermetallic compounds and Ir–Al intermetallics in the porous layer, as well as to confirm the presence or absence of elemental Ni and Ir and their phases after the process.

### 2.3. Electrochemical Measurements

Electrocatalytic performance for hydrogen evolution and urea oxidation was evaluated in a three-electrode electrochemical cell. The electrolyte was 1 M NaOH (aqueous) at room temperature. In tests involving UOR, 0.33 M urea was added to the NaOH solution (this concentration of urea is representative of urea-rich wastewater and is commonly used in UOR studies). All experiments used a saturated calomel electrode (SCE) as the reference electrode and a platinum mesh as the counter electrode. The working electrodes were the Ni-based samples described above (either smooth Ni, porous Ni, porous Ni–Ir, or porous Ni–Pt), with an exposed geometric area of 1.5 cm^2^.

Cyclic voltammetry (CV) was carried out to probe the general redox behavior and the activity for HER and UOR. Each electrode was cycled typically between –1.4 V and +0.8 V vs. SCE at a scan rate of 50 mV/s. This potential window encompasses the hydrogen evolution region on the negative end (around –1.4 V is where hydrogen evolution starts on Ni in alkaline media) and the urea oxidation/oxygen evolution region on the positive end (OER typically occurs above +0.5 V vs. SCE on Ni in 1 M NaOH, and UOR onset is around +0.37–0.4 V vs. SCE under these conditions). The CVs were recorded in the presence of 0.33 M urea to directly observe UOR activity, though, in some cases, baseline CVs in plain NaOH were also noted for reference. Current densities in all electrochemical data were normalized to the electrode’s geometric area for comparison.

Chronoamperometry (CA) tests were performed to evaluate the steady-state activity and stability of the electrodes for the UOR. For these tests, the electrode potential was held at a constant value of +0.40 V vs. SCE (which is near the onset of the urea oxidation peak in CV) for a duration of 60 min. The current was recorded as a function of time at this fixed potential. This allowed us to monitor how the UOR current evolved—whether it decays due to catalyst deactivation or remains stable—and how the different electrodes compare in sustaining the reaction. Prior to the chronoamperometry run, each electrode was typically pre-activated by a few CV cycles in the urea-containing electrolyte to ensure a stable surface state. All electrochemical experiments were performed at ambient temperature (~25 °C).

## 3. Results and Discussion

In Figure 1, SEM–EDS shows elemental mapping of the Ni and Ir deposit on the electrode surface. The deposition process was performed in mild electrochemical conditions (5 mA cm^2^) for a time of 60 min, which ensures good adhesion and avoids side reactions like hydrogen evolution. Ir forms a continuous metallic layer across the Ni surface. The obtained layer has around 7 micrometers of thickness and creates a smooth interface with conductive resin, which was used for cross-section preparation. This bilayer structure was used as a working electrode for the molten-salt modification process. Electrodes were immersed in a molten-salt mixture consisting of Al, which was potentiostatically deposited on the electrode surface at a potential of −1.4 V. According to our previous works related to different stainless steels, Ni, and Ni-Pt electrodes modified by Al deposition, a high temperature of the melt during the reduction of Al on the surface allows for the immediate formation of intermetallic phases. After a certain deposition time (1 h), reverse polarization takes place at a potential of −0.5 V, and deposited Al and Al-based compounds are electrochemically dissolved, which forms a porous, uniform structure.

The once smooth Ni/Ir surface is now characterized by an open network of pores and interconnected ligaments. Micron-scale voids and channels are visible alongside finer sub-micron porosity and crack-like features, all of which result from the volumetric expansion and Kirkendall void formation during Ni–Al alloying and Al dealloying. This structurally evolved Ni–Ir surface exhibits markedly increased surface area and a high density of low-coordinated defect sites, which should be beneficial for electrocatalytic reactions. The porous structure in Figure 2 is similar to that obtained in our previous Ni electrodes and Ni-Pt systems, indicating that adding Ir did not impede the alloying–dealloying process. 

Liquid Al penetrating the Ni (and Ni–Ir) at a high temperature reacts to form Ni–Al intermetallics, and upon Al removal, leaves behind a sponge-like Ni-rich matrix. Notably, the presence of Ir does not disrupt the formation of this porous “Raney-type” structure. Instead, Ir may subtly influence the microstructure—for example, by slightly refining the pore size or distribution—due to its effects on diffusion kinetics (Figure 3). These defects (microvoids and microcracks) serve as highly active sites for electrochemical reactions, as reported for selectively dealloyed materials, where increased defect density correlates with enhanced activity.

To examine the through-thickness distribution of elements, cross-sectional SEM/EDS analysis was performed on the porous Ni–Ir electrode. The Ni signal flows from the bulk substrate (bottom) through the porous layer (top), forming the continuous matrix of the electrode. Ir is detected across the entire modified layer, from the surface deep into the pores, indicating that Ir remains well-dispersed throughout the porous structure after aluminization and dealloying. Only a very faint Al signal is observed in the porous region, evidencing that the majority of Al introduced during the molten-salt treatment was removed during the dealloying step—any residual Al is minimal and likely bound in stable Ni–Al phases that resisted complete leaching. This compositional profile confirms that the porous layer is primarily composed of Ni (with embedded Ir), and that Ir incorporation is maintained uniformly across the thickness of the catalytically active layer X-ray diffraction was used to identify the crystalline phases present in the Ni–Ir electrodes before and after the molten-salt modification, as presented in Figure 4.

Before any treatment, the Ni sample shows diffraction peaks at 44.5°, 51.8°, and 76.4° 2θ, corresponding to the (111), (200), and (220) planes of face-centered cubic Ni. After Ir plating, some additional signals from the Ir metallic phase at 43.79 (111), 47.41 (200), and 73.72 (220) were detected. The intensity and location of Ir-based signals suggest that the deposited Ir has a nanostructural/amorphous structure with strong texture preference, due to the fact that, in the literature, the signals for (111) and (220) are on slightly different positions—40.83 and 76.20. After the molten-salt Al treatment and subsequent dealloying, the XRD pattern still exhibits the primary Ni (111), (200), and (220) peaks, but new minor peaks emerge at approximately 35.0°, 43.3°, and 57.5° 2θ. These additional reflections can be indexed to Ni3Al, with the (110), (111), and (210) planes, respectively. The presence of Ni_3_Al indicates that during the high-temperature alloying, Ni and Al formed intermetallic compounds, and while most of these compounds were leached out during dealloying, a small fraction (rich in Ni) remained undissolved. What should be noted is that signals from crystalline Ir signals are resolved, implying that Ir is well-integrated within the Ni-based phases. The electrochemical activity of the porous Ni–Ir electrodes was evaluated by cyclic voltammetry (CV) in an alkaline solution, and the results are compared to a baseline Ni electrode in Figure 5.

Baselines in electrochemical reactions in the tested system were established by a flat unmodified Ni electrode (red line). The cathodic part of the scan stops at −1.4 V and reaches around −18 mA cm^−2^. In terms of the anodic part, the potential for urea electro-oxidation was recorded at 0.4 V. Further scans show a gradual current increase and oxygen evolution reaction. Similar behavior has been observed in our previous work related to urea electro-oxidation on porous electrodes. A blue scan for porous Ni electrodes shows significant improvement in the catalytic activity of HER, reaching a 56 mA cm^−2^ current density for −1.4 V, and also shows excellent performance in urea electro-oxidation at 0.4 V (18 mA cm^−2^). Surprisingly, the Ni-Ir system was almost identical in terms of registered current density values in HER and OER, which was not expected, due to the fact that, in our previous studies in the Ni-Pt system, the presence of noble-metal addition significantly boosted overall electrochemical performance. These effects were only seen in this study for the Ni-Pt porous electrode (purple line), where cathodic current reached −100 mA cm^−2^ and anodic response at 0.4 V reached 20 mA cm^−2^. The presence of Pt and Ir can modify the catalytic chemistry of the surface. Ni–Ir in an anodic scan can achieve higher urea oxidation currents at lower overpotentials, improving efficiency, which was reported in many studies related to OER. This is a reason why we also tested these electrodes in the urea electro-oxidation reaction, and to our best knowledge, there are no reports related to porous Ni-Ir materials tested in UOR. The long-term performance and durability of the porous Ni–Ir electrodes were assessed via chronoamperometric measurements, as shown in Figure 6.

To further evaluate the electrocatalytic activity of the investigated electrodes in the urea oxidation reaction (UOR), chronoamperometric tests were performed at a constant potential of 0.40 V vs. SCE, which was selected on the basis of cyclic voltammetry results. At this potential, the anodic current is expected to originate predominantly from the UOR process. The chronoamperometric response of a smooth Ni electrode (Figure 6 purple line) revealed only a negligible initial current, which rapidly decayed to nearly zero during polarization. This clearly demonstrates the poor intrinsic activity of compact nickel surfaces toward UOR and their inability to sustain catalytic turnover over extended operations. A different behavior was observed for the porous Ni electrode fabricated by the molten-salt-assisted aluminum deposition/dissolution route. The initial current density at 0.40 V reached ~11 mA cm^−2^, and although a gradual decrease was noted with time, the electrode maintained a relatively high current density of ~9 mA cm^−2^ after 1 h of continuous polarization. This confirms that the enlarged electrochemically active surface area and the increased population of surface NiOOH/Ni(OH)_2_ redox sites play a crucial role in facilitating urea oxidation. Based on our previous experience with noble-metal incorporation into porous Ni matrices, we expected that the Ni–Ir electrode would exhibit further enhancement in UOR activity. Surprisingly, the initial current density was only ~7 mA cm^−2^, and within the first 5 min of polarization, it rapidly decayed to ~2 mA cm^−2^, with almost no recovery over the remainder of the experiment. Such behavior suggests that iridium, under anodic polarization in an alkaline medium, is prone to rapid surface oxidation to IrO_2_ or mixed Ir–Ni oxides, which may passivate the active Ni surface and hinder the formation of the Ni^2+^/Ni^3+^ redox couple. Since this redox couple is known to be a key intermediate in UOR, its suppression effectively diminishes the catalytic response of the Ni–Ir electrode. In contrast, the Ni–Pt electrode displayed an opposite trend. The initial current density was comparable to that of the porous Ni electrode (~11 mA cm^−2^); however, instead of decaying, the current gradually increased during the test and stabilized at ~11 mA cm^−2^ after 1 h. This sustained enhancement indicates a synergistic effect between Pt and Ni, where Pt sites facilitate urea adsorption and bond cleavage, while Ni redox transitions provide the active oxidative pathway. This cooperative action results in a stable catalytic performance superior to both porous Ni and Ni–Ir electrodes.

## 4. Conclusions

The molten-salt Al co-deposition and dealloying method produced a repeatable porous Ni layer and kept the pre-plated noble metal (Ir or Pt) inside the modified region. SEM/EDS and XRD showed a bicontinuous porous network mainly of fcc-Ni with only small Ni–Al remains. In UOR tests at +0.40 V vs. SCE (1 M NaOH + 0.33 M urea), smooth Ni was almost inactive. Porous Ni started at ~11 mA·cm^−2^, and after 60 min, it was ~9 mA·cm^−2^, which we link to the larger surface area and accessible Ni(II)/Ni(III) redox sites. Porous Ni–Ir started at ~7 mA·cm^−2^ but dropped to ~2 mA·cm^−2^ within 5 min, likely due to the fast formation of Ir oxides that block Ni sites and weaken the Ni redox couple. Porous Ni–Pt remained at ~11 mA·cm^−2^ over 60 min, suggesting a stable effect where Pt helps with urea adsorption/activation and Ni provides the redox path. In short, Pt addition improves UOR current and stability in this system, while Ir addition reduces them under the tested conditions.

## Figures and Tables

**Figure 1 materials-18-05069-f001:**
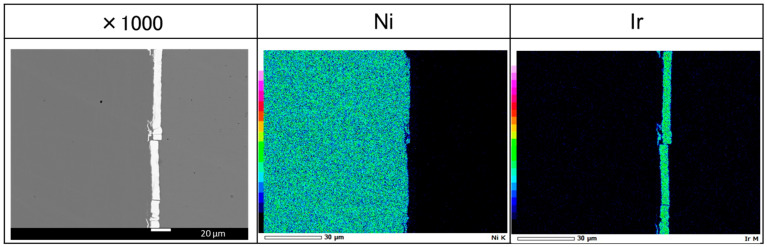
SEM observations of Ni electrode cross-section with plated Ir layer.

**Figure 2 materials-18-05069-f002:**
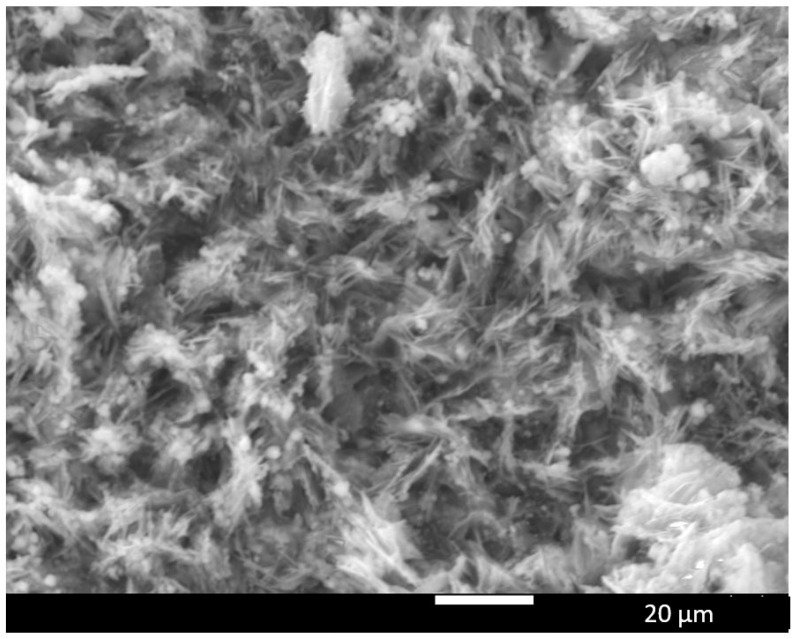
SEM micrographs of the Ni–Ir electrode after molten-salt Al co-deposition and dealloying at 1500× magnification.

**Figure 3 materials-18-05069-f003:**
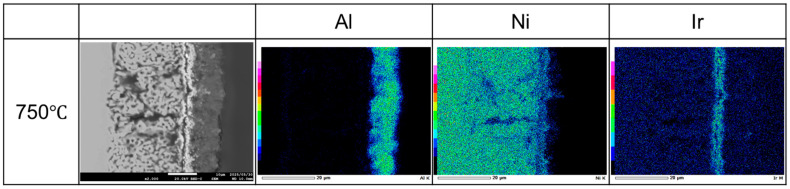
Cross-sectional SEM image of the Ni–Ir porous electrode with EDS elemental profiles for Al, Ni, and Ir.

**Figure 4 materials-18-05069-f004:**
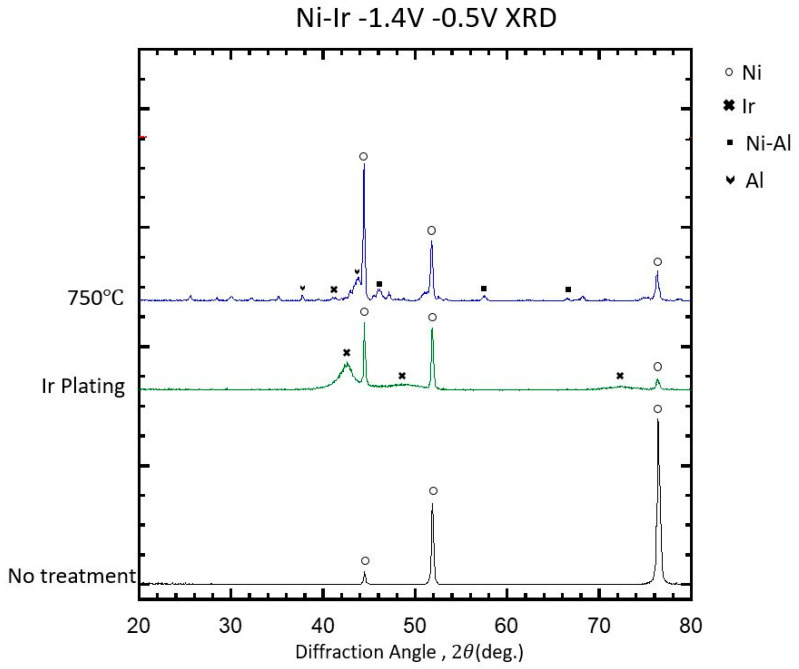
XRD patterns for the Ni layer, the Ni/Ir electrode, and the Ni–Ir electrode after Al dealloying.

**Figure 5 materials-18-05069-f005:**
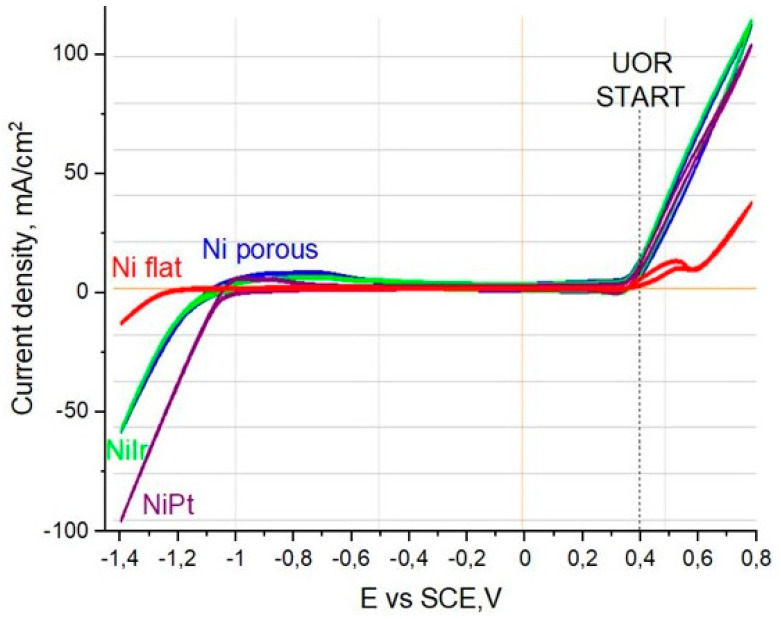
Cyclic voltammetry scans registered for a flat Ni electrode (red), porous Ni after treatment (blue), porous Ni-Ir (green), and porous Ni-Pt (purple) in a solution of 1 M NaOH + 0.33 M urea. Scan rate: 50 mV/s.

**Figure 6 materials-18-05069-f006:**
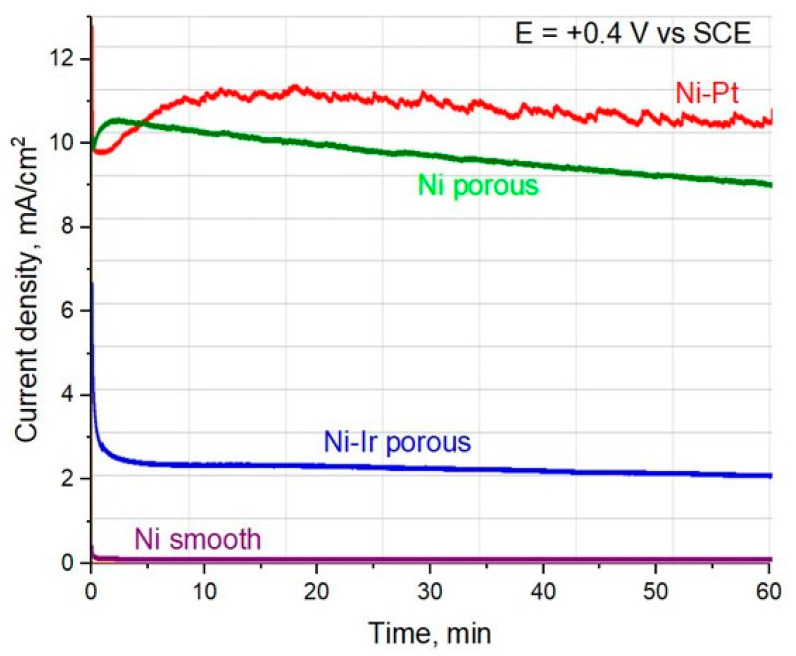
Time-dependent electrochemical behavior of Ni smooth (purple), Ni–Ir porous (blue), Ni porous (green), and Ni-Pt porous (red) electrodes. Chronoamperometry at a fixed anodic potential (e.g., +0.40 V vs. SCE) in 1 M KOH + 0.33 M urea.

## Data Availability

The original contributions presented in the study are included in the article, further inquiries can be directed to the corresponding authors.

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
