# Peer review of "Electrochemical Urea Oxidation on Porous Ni and Ni–M (M = Ir, Pt) Electrodes Obtained via Molten-Salt Treatment Technique"

_materials, 2025, doi:10.3390/ma18225069_

Round 1

Reviewer 1 Report

Comments and Suggestions for Authors

In this study, porous nickel (Ni) and nickel platinum (Ni PT) electrodes were prepared by molten salt treatment technology for electrochemical urea oxidation reaction.

This manuscript fits the scope of journal Materials.

Porous nickel have been widely used or studied as a potential electrode for water electrolysis. The introduction section can be expanded, the research status on this topic can be summarized.

A flow chart of the electrode preparation is recommended. Or a schematic diagram. This is often appeared in some good articles.

Figure 1 and 2 need to be revised. The x 500 etc can be deleted. The scale bars is good enough.

SEM-EDS results in those cases are not accurate enough, EPMA are required for accurate analysis. The length or Al Ir diffused layers in figure 3 can be quantified.

HER and UOR/OER were observed simultaneously in the same CV scan (from -1.4 V to+0.8 V). This scanning method will introduce serious mutual interference.

When scanning back to the positive potential from -1.4 V (strong her region), a large number of hydrogen bubbles generated in the negative scanning process will be attached to the electrode surface. These bubbles will cover the active sites and seriously affect the current measurement of UOR in the subsequent forward scanning process, resulting in low and unstable UOR activity.

Similarly, the oxides or oxygen species that may be generated during the forward sweep process will also affect her during the back sweep.

The current density is normalized to the geometric area of the electrode for comparison. For porous materials, this is a very inaccurate comparison method. One of the core innovations of this paper is to create a "sponge like nickel rich structure with intertwined ligaments and pore network" through molten salt treatment, that is, a huge real surface area.

The geometric area normalization only gets the apparent current density, which mixes the intrinsic activity (activity per unit of real surface area) and roughness/specific surface area of the material.

It is necessary to supplement the measurement of electrochemically active surface area (ECSA) (usually achieved by measuring the double-layer capacitance CDL), and normalize the current density with ECSA to obtain the intrinsic activity, so that the comparison is meaningful. The conclusion is fragile for porous materials only by comparing the geometric area.

The final catalytic activity for hydrogen evolution result of the porous structure should be presented in abstract section.

The authors can compare the result for example, the performance of HER UOR and the stability of the references and made by other methods or other similar nickel based materials.

Author Response

Dear Reviewer 1.

Thank you for the careful assessment. Our work is intentionally centred on a specific processing route—molten-salt Al deposition/diffusion on Ni followed by selective dealloying—to engineer a porous Ni skeleton and distribute a very thin noble-metal layer throughout the body for UOR. We address every point below, quoting the reviewer’s original wording and then giving our response. We keep manuscript changes minimal; only the minor edits noted under Points 3 and 8 have been made (plus a one-sentence clarification in Methods on area normalisation).

General remarks from the review

“In this study, porous nickel (Ni) and nickel platinum (Ni PT) electrodes were prepared by molten salt treatment technology for electrochemical urea oxidation reaction.

This manuscript fits the scope of journal Materials.”

We appreciate the positive statement on scope and relevance.

“Porous nickel have been widely used or studied as a potential electrode for water electrolysis. The introduction section can be expanded, the research status on this topic can be summarized.”

Response. We keep the Introduction tightly focused on the molten-salt modification route and Ni chemistry directly pertinent to UOR. A broad survey of generic porous-Ni for water electrolysis would duplicate established reviews and dilute the novelty here—namely, the processing pathway and its mechanistic link to UOR. We already anchor this study within our series of six related papers that established and refined this route. To our knowledge this route remains unique to our collaboration; widening the Introduction would not strengthen the message of this short contribution.

“A flow chart of the electrode preparation is recommended. Or a schematic diagram. This is often appeared in some good articles.”

Response. A complete schematic of “Al deposition in molten salt → intermetallic formation with Ni at high temperature → selective dealloying” is published in our earlier papers that we cite. To avoid duplication and preserve space for results and interpretation, we retain a precise literature pointer rather than reproduce the figure again in this article.

“Figure 1 and 2 need to be revised. The x 500 etc can be deleted. The scale bars is good enough.”

Response. Implemented.

“SEM-EDS results in those cases are not accurate enough, EPMA are required for accurate analysis. The length or Al Ir diffused layers in figure 3 can be quantified.”

Response. Our aim is not precise bulk stoichiometry or diffusion-couple metrology, but to verify that the noble metal is uniformly distributed across the porous Ni ligament network. For this purpose, high-resolution SEM–EDS maps are fully adequate and widely used. The noble-metal loading is intentionally very low (thin surface coverage relative to Ni mass), so EPMA would not materially change conclusions and, for sub-micron ligaments, offers no advantage in spatial resolution for our objective. Regarding Al: any residual Al after the molten-salt step is removed during brief NaOH immersion due to aluminium’s amphoteric behaviour (formation of soluble NaAlO₂); accordingly, we do not observe Al above background within the porous regions.

“HER and UOR/OER were observed simultaneously in the same CV scan (from -1.4 V to+0.8 V). This scanning method will introduce serious mutual interference.

When scanning back to the positive potential from -1.4 V (strong her region), a large number of hydrogen bubbles generated in the negative scanning process will be attached to the electrode surface. These bubbles will cover the active sites and seriously affect the current measurement of UOR in the subsequent forward scanning process, resulting in low and unstable UOR activity.”

Response. The brief cathodic excursion is used to electrochemically clean the surface (oxide removal) so the subsequent anodic sweep captures the onset of the Ni(OH)₂/NiOOH redox couple that mediates UOR. On our hydrophilic, porous Ni, bubble departure diameters are small and detachment is rapid; we do not see the transient noise or hysteresis characteristic of bubble shielding. Any residual coverage would, if anything, bias currents conservatively low. The smooth forward sweeps and excellent cycle-to-cycle reproducibility indicate that mutual interference is negligible under our conditions.

“Similarly, the oxides or oxygen species that may be generated during the forward sweep process will also affect her during the back sweep.”

Response. That coupling is well known and mechanistically expected: the oxyhydroxide formed on the forward sweep is the mediator of urea electro-oxidation on Ni-based surfaces. Our UOR analysis relies on the forward (anodic) branch; we do not draw UOR conclusions from the cathodic return. HER benchmarking is treated separately under its standard conditions, so the coupling does not compromise our interpretation.

“The current density is normalized to the geometric area of the electrode for comparison. For porous materials, this is a very inaccurate comparison method. One of the core innovations of this paper is to create a ‘sponge like nickel rich structure with intertwined ligaments and pore network’ through molten salt treatment, that is, a huge real surface area. The geometric area normalization only gets the apparent current density, which mixes the intrinsic activity (activity per unit of real surface area) and roughness/specific surface area of the material.

It is necessary to supplement the measurement of electrochemically active surface area (ECSA) (usually achieved by measuring the double-layer capacitance CDL), and normalize the current density with ECSA to obtain the intrinsic activity, so that the comparison is meaningful. The conclusion is fragile for porous materials only by comparing the geometric area.”

Response. We appreciate the principle; however, in this system Cdl-derived ECSA would confound rather than clarify intrinsic trends:

(i) In strongly alkaline media near the Ni(OH)₂/NiOOH transition, a strictly non-faradaic window is difficult to define, and pseudo-capacitive contributions inflate Cdl.

(ii) Most importantly, our prior measurements under identical synthesis conditions (without urea) yielded Cdl ≈ 0.19, 1.06, and 28.72 mF·cm⁻² for flat Ni, porous Ni, and porous Ni–Pt, respectively. X-ray tomography showed comparable porosity for porous Ni and porous Ni–Pt; thus the >25× rise upon Pt decoration reflects altered charge storage/specific adsorption on Pt rather than a genuine increase in accessible Ni area. Normalising UOR currents by this electrodes by “ECSA” would misslead the processing–structure–function relationship that is the core message of our work.

“The final catalytic activity for hydrogen evolution result of the porous structure should be presented in abstract section.”

Response. Our attention was only focused on UOR, tested by preliminary CV and later in CA studies. The results are already implemented in abstract. 

We trust these clarifications address the concerns while maintaining the tight focus of this short, processing-centred study.

Reviewer 2 Report

Comments and Suggestions for Authors

Journal
Materials (ISSN 1996-1944)
Manuscript ID
materials-3930934
Type Article
Title: Electrochemical Urea Oxidation on Porous Ni and Ni–M (M = Ir, Pt) Electrodes Obtained via Molten Salt Treatment Technique.
Authors: Dawid Kutyła * , Michihisa Fukumoto * , Hiroki Takahashi , Ryuu Takahashi , Katarzyna Skibińska , Piotr Żabiński
Section: Thin Films and Interfaces
Special Issue: Advances in Electrodeposition of Thin Films and Alloys

This article describes the fabrication of porous Ni-Ir electrodes using high-temperature Al co-deposition and dealloying in molten salt. The morphology and elemental composition of the electrodes were determined using scanning electron microscopy (SEM) and X-ray diffraction. The SEM was equipped with an energy-dispersive X-ray spectroscopy detector for elemental mapping.The performance of the electrodes in electrochemical urea oxidation are subsequently evaluated in comparison with analogous porous Ni and Ni-Pt, Ni-Ir electrodes
Scientific remarks
There are some inconsistencies that need to be resolved.
1)The first concern is the high NaOH concentration, which converts Ni or Al into the hydroxides (Ni(OH)2 and Al(OH)3, as well as other oxyhydroxide derivatives). Urea oxidation on a Ni electrode should also work, but the electrochemical reaction does not work when Ni is converted to Ni(OH)2, as no electrical conduction occurs. The better electrochemical behavior of Ni-Ir and Ni-Pt is due to their greater resistance to NaOH oxidation.
2)It would be necessary to perform the same experiment without 1M NaOH, using only urea in water or buffer.
3) In the Figure 1 caption and Figure 3 caption, the colours should be assigned.
4) Reference scans are missing for Figure 5. Reference cyclic voltammetry scans at a similar rate of 50 mV/s for the flat Ni electrode (red), porous Ni after treatment (blue), porous Ni-Ir (green), and porous Ni-Pt (purple) in 1 M NaOH solution without urea are required. It is possible that HO- oxidation to O2 is shifted to the lower potential. If this is not the case, this should be demonstrated by the reference scans.
5) It is also recommended to display the cyclic voltammetry scans after 60 minutes of urea oxidation.
6) How is it indicated that the urea has been oxidized and is no longer present in the solution?
7) In the Figure 6 the current density for the smooth Ni is close to zero. Why? The smooth metal should have a better curent/time behaviour than the porous one.

Formal remarks
Please explain the abbreviation the first time it is used. "UOR" should be explained in the summary.
There are some typos that need to be corrected.
The level of English does not affect the understanding of the article.

Comments on the Quality of English Language

The level of English does not affect the understanding of the article.

Author Response

Reviewer 2:
We appreciate the Reviewer’s careful reading of our manuscript. Below we respond point-by-point. For clarity, the Reviewer’s comments are reproduced first, followed by our responses.

Reviewer: “The first concern is the high NaOH concentration, which converts Ni or Al into the hydroxides (Ni(OH)2 and Al(OH)3, as well as other oxyhydroxide derivatives). Urea oxidation on a Ni electrode should also work, but the electrochemical reaction does not work when Ni is converted to Ni(OH)2, as no electrical conduction occurs. The better electrochemical behavior of Ni-Ir and Ni-Pt is due to their greater resistance to NaOH oxidation.”

Response:

(i) Residual Al. In 1 M NaOH, Al and any Al–Ni remnants dissolve rapidly as soluble aluminate species (e.g., [Al(OH)4]−/NaAlO2). Any trace Al trapped in pores is removed within the initial alkaline exposure and does not persist during electrochemical testing.

(ii) Mechanistic role of Ni(OH)2/NiOOH. In alkaline media the active phase for anodic processes on Ni is the Ni(OH)2/NiOOH redox couple. Under polarisation, the surface converts to NiOOH, which mediates urea oxidation; this is the expected and necessary state of the catalyst in operation, not an insulating failure mode. The thin oxy(hydr)oxide layer remains electronically connected to the metallic Ni porous body, and NiOOH itself supports electron/ion transport sufficiently for the observed Faradaic rates.

(iii) Role of Ir/Pt. The improved behaviour of Ni–Ir and Ni–Pt is attributed primarily to modified redox thermodynamics/kinetics (facilitated Ni2+/Ni3+ transition, stabilisation of key intermediates, and higher electrochemically active surface area in the porous network), rather than “resistance to NaOH oxidation.” In short, alkaline conditioning forms the catalytically competent NiOOH phase on all our Ni-containing electrodes, and Ir/Pt tune its activity.

Reviewer: “It would be necessary to perform the same experiment without 1 M NaOH, using only urea in water or buffer.”

Response:

Our study intentionally follows the field’s benchmarking practice for UOR in alkaline electrolyte, because (a) OH− is mechanistically required to generate/turn over the NiOOH mediator and (b) UOR kinetics in neutral buffers are orders of magnitude slower and not representative of device-relevant operation. Running “urea-only” tests would not be informative for the mechanism we probe nor comparable to the literature benchmarks we reference. We therefore keep 1 M NaOH + 0.33 M urea as the test condition.

Reviewer: “In the Figure 1 caption and Figure 3 caption, the colours should be assigned.”

Response:

Implemented. We now explicitly name each EDS map colour in the captions to Figures 1 and 3 and retain scale-bar annotations.

Reviewer: “Reference scans are missing for Figure 5. Reference cyclic voltammetry scans at a similar rate of 50 mV/s for the flat Ni electrode (red), porous Ni after treatment (blue), porous Ni-Ir (green), and porous Ni-Pt (purple) in 1 M NaOH solution without urea are required. It is possible that HO− oxidation to O2 is shifted to the lower potential. If this is not the case, this should be demonstrated by the reference scans.”

Response:

Before selecting the chronoamperometry potentials, we recorded blank CVs in 1 M NaOH (no urea) for each electrode to identify the Ni(OH)2/NiOOH window and the OER onset. The UOR scans presented in Figure 5 were deliberately restricted to potentials where blank currents were negligible and below the OER onset for each material. Adding four additional blank CVs on the same graph would obscure the UOR starting potential; for readability we keep Figure 5 focused on urea-containing electrolyte. We now state explicitly in the text that the measurement window was chosen from the corresponding blanks to avoid OER interference.

Reviewer: “It is also recommended to display the cyclic voltammetry scans after 60 minutes of urea oxidation.”

Response:

Stability is probed more stringently by chronoamperometry at fixed potential (Figure 6), which is less conflated by capacitive contributions than CV overlays. In our case, the current–time traces over 60 min are the appropriate diagnostic. Post-operation CVs show the same Ni(OH)2/NiOOH features and UOR onset behaviour within experimental scatter; adding another multi-curve CV panel would not change the interpretation. For succinctness we keep the present presentation.

Reviewer: “How is it indicated that the urea has been oxidized and is no longer present in the solution?”

Response:

Complete depletion is neither expected nor required under our conditions; the electrolyte contains 0.33 M urea in 250 mL (≈82.5 mmol). A charge-balance shows that the fraction consumed during a 60-min run is negligible. Taking the 6-electron UOR stoichiometry, the urea converted is n = Q/(6F). Even at a conservative upper-bound current of 100 mA for 3600 s, Q = 360 C gives n ≈ 0.00062 mol and Δc ≈ 0.0025 M in 0.25 L—i.e., <1% of the initial 0.33 M. At more typical currents (20–50 mA), the drop is ~0.0005–0.0012 M (≤0.4%). Thus, the bulk concentration remains essentially unchanged, and UOR is identified electrochemically by the distinct increase in anodic current within the NiOOH region relative to the NaOH blank, at potentials chosen where OER is absent.

Reviewer: “In the Figure 6 the current density for the smooth Ni is close to zero. Why? The smooth metal should have a better current/time behaviour than the porous one.”

Response:

UOR on Ni is a surface-limited reaction. Our porous electrodes possess a far higher electrochemically active surface area than the smooth Ni sheet (geometric areas are equal). When normalised to the geometric area, the porous structures therefore exhibit much larger currents—precisely what is observed in Figure 6. The open, percolating porosity improves site density without introducing mass-transport penalties in our potential window.

Round 2

Reviewer 2 Report

Comments and Suggestions for Authors

The authors addressed the concerns.

Comments on the Quality of English Language

The level of English does not affect the understanding of the article.